# Heavy Metals/Metalloids in Soil of a Uranium Tailings Pond in Northwest China: Distribution and Relationship with Soil Physicochemical Properties and Radionuclides

Yu Mao [1], Jinlong Yong [1], Qian Liu [2], Baoshan Wu [1], Henglei Chen [1], Youhua Hu [3] and Guangwen Feng [1],*

[1] Research Center of Radiation Ecology and Ion Beam Biotechnology, College of Physics Science and Technology, Xinjiang University, Urumqi 830017, China; maoyu473380219@163.com (Y.M.); yongjinlong32@163.com (J.Y.); wubs@xju.edu.cn (B.W.); chl19790413@163.com (H.C.)

[2] School of Statistics and Data Science, Xinjiang University of Finance & Economics, Urumqi 830012, China; liuqian121314@163.com

[3] Radiation Environment Supervision Station of Xinjiang, Urumqi 830000, China; hyh1229@126.com

* Correspondence: feng_guang_wen@163.com

**Abstract:** Uranium tailings ponds have a potential impact on the soil ecological environment and human health. In this study, the measurement and spatial distribution characteristics of soil physicochemical properties (pH, EC, TN, TOC, and TP) and heavy metals/metalloids (Cd, Pb, Zn, Cr, and As) in two different profiles (0–5 cm, 5–15 cm) were completed and visualized in a decommissioned uranium tailings pond in Northwest China. The results showed that almost all measured values in the study area were within the background values of China and other countries or regions around the world. The visual spatial distribution map showed that the spatial distribution characteristics of the EC, TP content, Pb content, and Cr content of the soil in the tailings pond and its adjacent area increased with the increase in depth of the vertical profile. The visual correlation heatmap analysis found that, in general, there were significant positive correlations among heavy metals and radionuclides and significant negative correlations among heavy metals, radionuclides, and physicochemical properties. The cluster tree divided environmental factors into two clusters; pH, TP, $^{40}$K, Cd, and Zn formed one cluster, which could be related to the similar structures and physicochemical properties of Cd and Zn, and Pb, Cr, $^{232}$Th, TN, EC, TOC, As, $^{238}$U, and $^{226}$Ra formed another cluster of lithophile elements with similar geochemical properties. Based on the analysis results, the uranium tailings pond is in good operation, and no migration and diffusion of heavy metals/metalloids to the surrounding soil ecological environment was found.

**Keywords:** uranium mining and milling; metal/metalloid; physicochemical properties; spatial distribution; correlation analysis

## 1. Introduction

With the rapid development of the global economy and sustained population growth, the energy demand is likewise increasing sharply. Nowadays, resulting from the considerable negative environmental factors caused by the supply and consumption of fossil fuel (such as coal, oil, and natural gas), combined with their insufficient storage as well as the restrictions on greenhouse gas emissions all over the world, there is increasing interest in nuclear energy known to be more efficient, clean and economical [1–4]. As a strategic resource, uranium is critical to improving the quality of the environment and adjusting the structure of energy, as well as promoting its application and development in military and civil fields [5–7]. It is, however, impossible to avoid generating a large volume of waste during the process of uranium mining and milling (e.g., overburden, mineralized wastes, residual wastes, etc.) [8–14]. As a rule, these waste materials still contain huge amounts of long-lived radionuclides, for example, uranium, thorium, radium and others, as well as

associated typical heavy metals (for example, Cd, Pb, Cr, etc.) and metalloids (for example, As) [7,9–11,15]. These radionuclides and heavy metals/metalloids may be influenced by natural or human factors and may migrate into environmental media, thus affecting the soil ecosystem, food chain, and human health around tailings ponds [9,10,15]. Researchers in Portugal, the UK, and China have carried out investigations on the distribution of radionuclides and heavy metals/metalloids in uranium mining areas. The results show that radionuclides and heavy metals/metalloids can not only affect the organs, tissues, and immune system of residents around the mining area, and even cause cancer, but can also lead to serious excessive heavy metals/metalloids in crops around the mining area [16–18]. For example, the excessive rate of As in rice planted around some uranium mining areas has reached 97.73% [18]. In addition, the extraction and recovery of radionuclides and heavy metals/metalloids from waste materials are also facing several constraints such as technic-economic, environmental, and social aspects [19]. Thus, a large number of uranium tailings ponds are built to bury the waste materials to prevent radionuclide and heavy metal/metalloid migration and diffusion to the surrounding environment media and maintain long-term stabilization [20–22]. Nevertheless, with the increasing number of service years of the tailings ponds and the existence of some adverse conditions such as natural or anthropogenic factors, the treated radionuclides and heavy metals/metalloids in the tailings pond still have potential risk of migration and diffusion to the adjacent environment and transfer to the food chain, affecting human health [5,19,23]. Therefore, in view of the characteristics of radionuclides and heavy metals/metalloids in the ecological environment, uranium tailings ponds have attracted extensive attention of researchers and governments all over the world [11,19,20,23–25].

In order to understand the remedial effect and operating status of a decommissioned uranium tailings pond in Northwest China and to prevent ecological environment pollution and public health risks, this research was carried out. In this study, based on a previous investigation of the radioactive level of the uranium tailings pond [19], an investigation of the soil physicochemical properties and the soil metals/metalloids was conducted firstly on the uranium tailings pond and its adjacent area, and then the spatial distribution of the soil physicochemical properties and the soil metals/metalloids was characterized. Finally, combined with the previously published soil radionuclide activity concentration [19], the correlation between the soil physicochemical properties, metals/metalloids, and radionuclides was analyzed. This work is important for monitoring and understanding the migration and diffusion of soil heavy metals/metalloids in uranium tailings ponds and their adjacent area.

## 2. Materials and Methods

### 2.1. Study Area

A uranium mining and milling tailings pond in Northwest China was decommissioned more than 10 years ago through an impervious layer, cement slope protection, topsoil covering, and other projects. The location was described in the figure of our previous study [19]. The study area where the uranium tailings pond (about 20,000 m$^2$) is located has an arid and semi-arid climate in the middle temperate zone, with the following conditions: perennial drought and rainless, strong sunshine, sparse surface vegetation, and an underdeveloped surface water system. The annual average temperature in this area is 6.5 °C, and the average altitude is 926 m. The overall terrain of the uranium tailings pond is characterized by high in the north, low in the south, high in the west, and low in the east. The northeastern area of the uranium tailings pond is under concrete slope protection, and the southeastern area consists of hills [19].

### 2.2. Sampling and Sample Pretreatment

The samples for measuring physicochemical properties and the heavy metal/metalloid content of the soil were collected from two vertical depth profiles, 0–5 cm and 5–15 cm, in the uranium tailings pond and its adjacent area. The 14 soil sampling points are shown

in Figure 1, which are basically consistent with those of soil radionuclide measurement sample [19]. As shown in Figure 1, "T" represents the internal area of the uranium tailings pond, "C" represents the adjacent area around the tailings pond, "S" represents the sampling depth of 0–5 cm, and "X" represents the sampling depth of 5–15 cm. The samples at depths of 0–5 cm in the uranium tailings pond are named TS1, TS2, TS3, and TS4, and those at depths 5–15 cm in the uranium tailings pond are named TX1, TX2, TX3, and TX4. The three sampling points in the adjacent area of the tailings pond are named CS1, CS2, and CS3 at depths 0–5 cm, and samples at depths 5–15 cm are named CX1, CX2, and CX3. For more sampling details, refer to our previous study [19]. The mixed samples processed at each sampling point were divided into three parts. One part was placed in a premarked sampling bag for the analysis of soil radionuclides, one part was placed in a premarked sampling bag for the measurement of soil physicochemical properties and heavy metals/metalloids, and the remaining part was used for the measurement of other items.

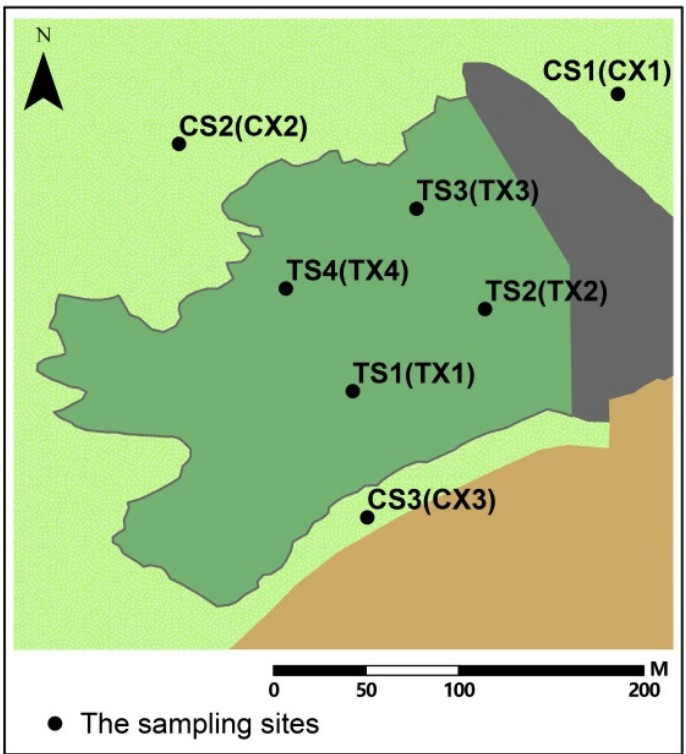

**Figure 1.** The soil sampling sites.

The pretreatment of the soil samples was carried out following the corresponding measurement and analysis standards. Impurities such as gravel and debris were removed through a sieve, and then the soil samples were dried in a blast drying oven at 105 °C to a constant weight. The dried soil samples were ground into a fine powder with a test sieve; the powder samples that passed through a 2 mm sieve were used for determination of soil pH and electrical conductivity (EC), while those that passed through a 0.15 mm sieve were used for soil total nitrogen (TN), total organic carbon (TOC), total phosphorus (TP), and heavy metal/metalloid analysis.

### 2.3. Analytical Methods

Analyses of the physicochemical properties mainly included the soil pH, electrical conductivity (EC), total nitrogen (TN), total organic carbon (TOC), and total phosphorus (TP). Soil pH was determined using a pH meter (Model PHBJ-261L, Shanghai INES Scientific Instrument Co., Ltd., Shanghai, China) in a soil/water suspension (soil liquid ratio

1:2.5 mL/g) [26], while soil organic carbon (SOC) was measured using the potassium dichromate oxidation method, using an oil bath (Model DF-101S, Shanghai Lichen Instrument Technology Co., Ltd., Shanghai, China), temperature-regulating electric furnace (Model DL-01, Beijing Yongguang Medical Instrument, Beijing, China), and 250 °C thermometer (Model WNG-01, Changzhou Jiuhong Instrument Co., Ltd., Changzhou, China) [27]. The electrical conductivity (EC, μS/cm) of soils was measured using an electrode method, determined by a conductivity meter (Model DDBJ-351L, Shanghai INES Scientific Instrument Co., Ltd., Shanghai, China) [28]. Total nitrogen (TN, g/kg) and total phosphorus (TP, g/kg) in soil were measured using the methods recommended in the literature, i.e., an automatic Kjeldahl apparatus (Model KDY-9820, Beijing Tongrunyuan Electromechanical Technology Co., Ltd., Beijing, China) and ultraviolet–visible spectrophotometer (Model 754, Shanghai Sunny Hengping Instrument Co., Ltd., Shanghai, China), respectively [29].

The activity concentrations of natural radionuclides in each soil sample were analyzed by gamma-ray spectroscopy [30]. Detailed gamma-ray spectroscopy measurements of soil samples can be found in our previous study [19].

The determination of soil heavy metals/metalloids included the measurement of the content of heavy metals such as cadmium (Cd), lead (Pb), zinc (Zn), and chromium (Cr), as well as metalloid arsenic (As) in the soil. These selected elements in soil samples after strong acid digestion were measured using the methods recommended in the literature, determined by an inductively coupled plasma optical emission spectrometer (Model Optima 8000, Perkin Elmer, Waltham, MA, USA) [31].

All chemical reagents used in the analytical method were analytically pure or superior pure. The replicates of 10% of the total samples and blank corrections were also analyzed simultaneously to ensure the accuracy of the analytical results.

### 2.4. Statistical Analysis and Data Processing

The measured data were processed using IBM SPSS 21.0 Statistics software for Windows [32] and were statistically analyzed by R software version 3.6.1 for Windows [33]. Many user-written packages with different functions together constitute the powerful R software [34]. The psych package and pheatmap package of R software were used to complete the correlation analysis and heat map drawing, the ggplot2 package of R software was used to visualize the data, and ArcGIS 10.2 software for Windows was used to complete the inverse distance weighting analysis and visualization [33,35].

## 3. Result and Discussion

### 3.1. Soil Physicochemical Properties

The physicochemical properties of soil samples in different vertical profiles of the uranium tailings pond and its adjacent area were determined, and these results are described in Table 1. In the profile (0–5 cm) of the uranium tailings pond, the measured ranges of pH, electrical conductivity (EC), total nitrogen (TN), total organic carbon (TOC), and total phosphorus (TP) in the soil samples were 7.79–8.75, 136.90–1115.00 μS/cm, 0.58–1.01 g/kg, 3.18–15.20 g/kg, and 0.27–0.75 g/kg, respectively, while in the profile (5–15 cm) of uranium tailings pond, the values were 7.92–8.15, 112.20–1017.00 μS/cm, 0.56–1.44 g/kg, 1.88–16.50 g/kg, and 0.27–0.69 g/kg, respectively. In the profile (0–5 cm) of the adjacent area, the measured values of pH, electrical conductivity (EC), total nitrogen (TN), total organic carbon (TOC), and total phosphorus (TP) in soil samples were 8.03–9.23, 165.50–481.00 μS/cm, 0.48–0.59 g/kg, 0.99–1.91 g/kg, and 0.36–0.63 g/kg, respectively, while in the profile (5–15 cm) around the uranium tailings pond, the values were 7.88–8.38, 560.00–1015.00 μS/cm, 0.47–0.58 g/kg, 0.32–2.12 g/kg, and 0.30–0.95 g/kg, respectively.

**Table 1.** The soil physicochemical properties in the uranium tailings pond and its adjacent area.

| Sites | Physicochemical Properties | Statistical Parameters | Vertical Profile Depth (cm) | |
|---|---|---|---|---|
| | | | 0–5 | 5–15 |
| Inside the tailings pond | pH | Range | 7.79–8.75 | 7.92–8.15 |
| | | Mean ± SD | 8.09 ± 0.45 | 8.03 ± 0.10 |
| | EC (μS/cm) | Range | 136.90–1115.00 | 112.20–1017.00 |
| | | Mean ± SD | 843.98 ± 473.88 | 709.80 ± 405.89 |
| | TN (g/kg) | Range | 0.58–1.01 | 0.56–1.44 |
| | | Mean ± SD | 0.73 ± 0.20 | 0.79 ± 0.43 |
| | TOC (g/kg) | Range | 3.18–15.20 | 1.88–16.50 |
| | | Mean ± SD | 9.03 ± 6.49 | 6.13 ± 6.99 |
| | TP (g/kg) | Range | 0.27–0.75 | 0.27–0.69 |
| | | Mean ± SD | 0.48 ± 0.24 | 0.43 ± 0.20 |
| Adjacent area of the tailings pond | pH | Range | 8.03–9.23 | 7.88–8.38 |
| | | Mean ± SD | 8.50 ± 0.64 | 8.11 ± 0.25 |
| | EC (μS/cm) | Range | 165.50–481.00 | 560.00–1015.00 |
| | | Mean ± SD | 347.83 ± 163.40 | 863.33 ± 262.69 |
| | TN (g/kg) | Range | 0.48–0.59 | 0.47–0.58 |
| | | Mean ± SD | 0.54 ± 0.06 | 0.54 ± 0.06 |
| | TOC (g/kg) | Range | 0.99–1.91 | 0.32–2.12 |
| | | Mean ± SD | 1.53 ± 0.48 | 1.11 ± 0.92 |
| | TP (g/kg) | Range | 0.36–0.63 | 0.30–0.95 |
| | | Mean ± SD | 0.49 ± 0.14 | 0.68 ± 0.34 |

The measured values of pH in the study area were within the range of the domestic soil background value of pH: 3.10–10.60 [31], which further showed that the soil in the study area was alkalescence to alkaline, while the values of electrical conductivity (EC) in the study area were relatively high, which could be related to the arid and semi-arid climate in the study area, and the measured values were consistent with the relevant reports on the electrical conductivity (EC) of topsoil in arid and semi-arid areas and also reflected the high salinity of the soil in the study area to a certain extent [36,37]. Compared with the topsoil survey report of the European Commission (LUCAS report, 2013) (pH: <4.5–9.0, TN: <0.5–3.0 g/kg, TOC: <10–120 g/kg and TP: <10–80 mg/kg) [38], the measured values of pH and total nitrogen (TN) in the study area were within the range reported, while the values of total organic carbon (TOC) in the study area were lower, which could be related to the poor soil and scarce vegetation in the study area, and the values of total phosphorus (TP) in the study area were higher, which could be due to the drought and lack of rainfall in the study area that reduced the leaching of phosphorus from the soil.

A comparison of soil physicochemical properties in the uranium tailings pond and its adjacent area is shown in Figure 2A. The soil physicochemical properties in the uranium tailings pond and its adjacent area showed no significant difference except for the TN and TOC contents based on the Wilcoxon rank sum test. The reason that the TN and TOC contents in soil in the uranium tailings pond were higher than those outside the pond could be as follows: (1) The decommissioning time of the uranium tailings pond was relatively short, and it took time for indigenous microorganisms to recover to the undisturbed state, which could lead to lower microbial diversity inside than outside, affecting the decomposition of the TN and TOC in the uranium tailings. (2) The vegetation in the uranium tailings was richer than that outside, which increased the storage of the TN and TOC in the soil.

### 3.2. Soil Metal/Metalloid Content Determination

The heavy metal/metalloid contents of soil samples in different vertical profiles of the tailings pond and its adjacent area are shown in Table 2. In the profile (0–5 cm) of the uranium tailings pond, the measured ranges of As, Cd, Pb, Zn, and Cr in soil samples were 119.33–124.15 μg/g, 0.15–0.68 μg/g, 11.51–26.70 μg/g, 145.60–159.97 μg/g, and 19.75–38.88 μg/g, respectively, while in the profile (5–15 cm) of the uranium tailings pond, the values were 118.25–139.80 μg/g, 0.22–0.31 μg/g, 16.13–27.21 μg/g, 140.95–157.23 μg/g, and 24.91–35.928 μg/g,

respectively. In the profile (0–5 cm) of the adjacent area, the measured ranges of As, Cd, Pb, Zn, and Cr in soil samples were 124.35–152.59 µg/g, 0.23–1.02 µg/g, 10.05–30.94 µg/g, 396.99–544.20 µg/g, and 16.62–42.71 µg/g, respectively, while in the profile (5–15 cm) of the uranium tailings pond, the values were 88.10–234.59 µg/g, 4.79–36.64 µg/g, 157.76–394.84 µg/g, 13.45–45.17 µg/g, and 24.91–35.928 µg/g, respectively.

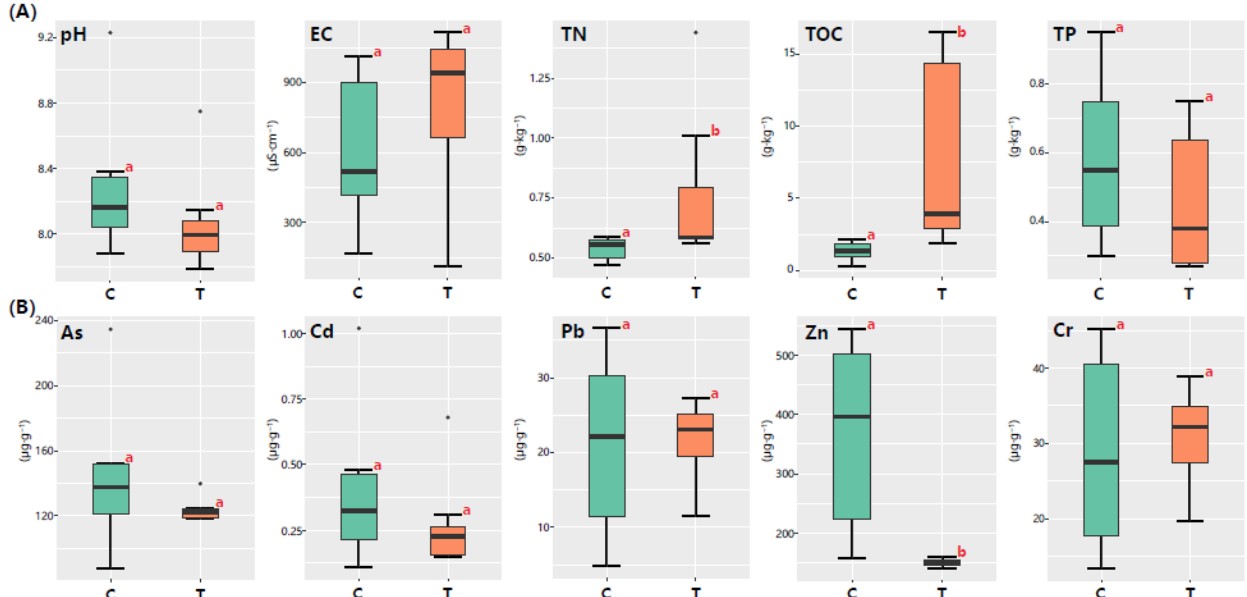

**Figure 2.** Physicochemical properties (**A**) and heavy metal/metalloid contents (**B**) of soil in the uranium tailings pond (T) and its adjacent area (C). Different lowercase letters indicate a significant difference between groups ($p < 0.05$) (Wilcoxon rank sum test).

**Table 2.** Heavy metal/metalloid contents of soil samples in the uranium tailings pond and its adjacent area.

| Sites | Metal/Metalloid (µg/g) | Statistical Parameters | Two Profile Depth (cm) | |
|---|---|---|---|---|
| | | | 0–5 | 5–15 |
| Inside the tailings pond | As | Range | 119.33–124.15 | 118.25–139.80 |
| | | Mean $\pm$ SD | 122.13 $\pm$ 2.28 | 125.32 $\pm$ 10.13 |
| | Cd | Range | 0.15–0.68 | 0.22–0.31 |
| | | Mean $\pm$ SD | 0.29 $\pm$ 0.26 | 0.25 $\pm$ 0.04 |
| | Pb | Range | 11.51–26.70 | 16.13–27.21 |
| | | Mean $\pm$ SD | 20.69 $\pm$ 6.62 | 22.57 $\pm$ 4.75 |
| | Zn | Range | 145.60–159.97 | 140.95–157.23 |
| | | Mean $\pm$ SD | 150.78 $\pm$ 6.44 | 148.49 $\pm$ 8.30 |
| | Cr | Range | 19.75–38.88 | 24.91–35.928 |
| | | Mean $\pm$ SD | 30.36 $\pm$ 8.31 | 31.31 $\pm$ 4.68 |
| Adjacent area of the tailings pond | As | Range | 124.35–152.59 | 88.10–234.59 |
| | | Mean $\pm$ SD | 142.65 $\pm$ 15.87 | 147.69 $\pm$ 76.97 |
| | Cd | Range | 0.23–1.02 | 0.11–0.48 |
| | | Mean $\pm$ SD | 0.56 $\pm$ 0.41 | 0.27 $\pm$ 0.19 |
| | Pb | Range | 10.05–30.94 | 4.79–36.64 |
| | | Mean $\pm$ SD | 18.19 $\pm$ 10.81 | 23.34 $\pm$ 16.56 |
| | Zn | Range | 396.99–544.20 | 157.76–394.84 |
| | | Mean $\pm$ SD | 493.18 $\pm$ 83.35 | 239.83 $\pm$ 134.32 |
| | Cr | Range | 16.62–42.71 | 13.45–45.17 |
| | | Mean $\pm$ SD | 26.81 $\pm$ 13.95 | 30.85 $\pm$ 16.08 |

A comparison of the heavy metal/metalloid contents of soil in the uranium tailings pond and its adjacent area is shown in Figure 2B. The soil physicochemical properties in the

uranium tailings pond and its adjacent area showed no significant difference except for the Zn content based on Wilcoxon rank sum test. The reason that the Zn content of soil in the uranium tailings pond was lower than that outside could be as follows: (1) The impervious layer was used in the construction of the uranium tailings pond, which prevented the migration of Zn in soil from a high to a low concentration. (2) The vegetation in the uranium tailings was richer than that outside, and the Zn in the soil could be absorbed during vegetation growth [39].

The background heavy metal/metalloid values of soil in several countries or regions around the world are shown in Table 3. The measured values of different heavy metals/metalloids in the soil of the uranium tailings pond and its adjacent area were within the range of background values of soil elements in China and other countries or regions around the world (As: <0.1–626.0 µg/g, Cd: 0.001–13.5 µg/g, Pb: 0.68–1143.0 µg/g, Zn: 2.5–2900 µg/g, Cr: 0.20–3000 µg/g) [31,40–45].

**Table 3.** Comparison of the heavy metal/metalloid contents of soil in several countries or regions around the world.

| Countries/Regions | Statistical Parameter | Heavy Metal/Metalloid Content (µg/g) | | | | | References |
|---|---|---|---|---|---|---|---|
| | | As | Cd | Pb | Zn | Cr | |
| United States | | <0.1–97 | - | <10–700 | <5–2900 | 1–2000 | [40] |
| Japan | | - | 0.021–3.4 | 1.0–1100 | 2.5–330 | 1.4–230 | [41] |
| Italy | | 4–197 | 0.07–0.80 | 4–81 | 16–157 | 20–307 | [42] |
| European Union | Range | 0.46–252.53 | 0.02–3.17 | 1.6–151.12 | - | 1.57–273.94 | [43] |
| Branicevo, Serbia | | - | <0.01–13.5 | 3.17–192 | 1.82–303 | 0.20–142 | [44] |
| Continental crust | | 0.1–40 | 0.01–0.7 | 2–100 | 10–300 | 5–3000 | [45] |
| China | | 0.01–626.0 | 0.001–13.4 | 0.68–1143.0 | 2.6–593.2 | 2.2–1209.0 | [31] |
| Study area, China | | 88.10–234.59 | 0.11–1.02 | 4.79–36.64 | 140.95–544.2 | 13.44–45.17 | Present study |

### 3.3. Soil Physicochemical Properties and Heavy Metal/Metalloid Distribution Characteristics

The inverse distance-weighted interpolation method is a spatial local weighted average interpolation method based on the principle of geographic similarity, which calculates the discrete points adjacent to the point to be measured for the weighted average [46]. Based on the measured data of soil physicochemical properties and heavy metal/metalloids, the spatial distribution characteristics of soil physicochemical properties (such as pH, EC, TN, TOC, and TP) and heavy metal/metalloids (such as As, Cd, Pb, Zn, and Cr) in different horizontal and vertical profiles are illustrated in Figures 2 and 3.

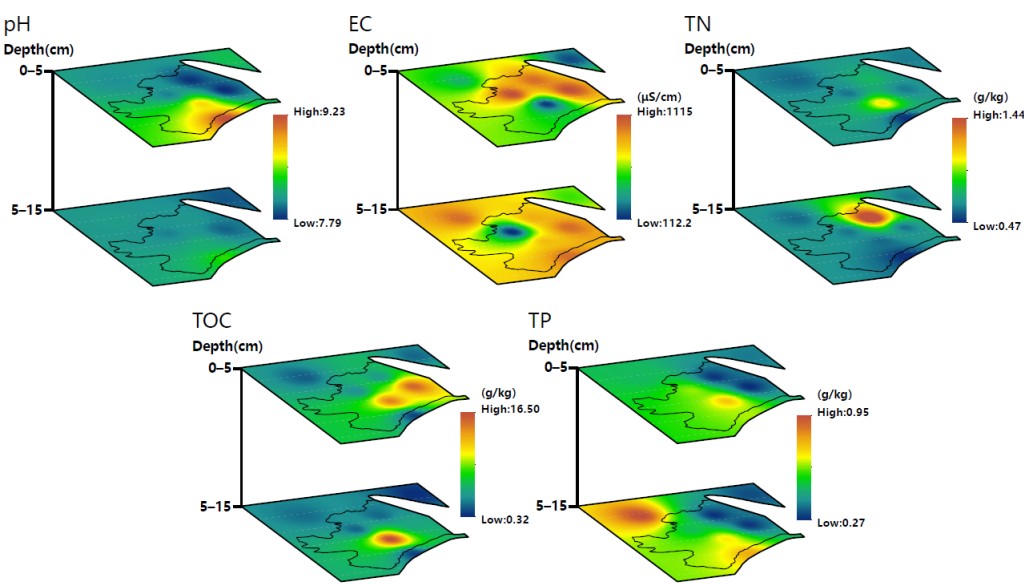

**Figure 3.** Spatial distribution characteristics of soil pH, EC, TN, TOC, and TP.

As shown in Figure 3, the soil pH in the horizontal profile of 0–5 cm was relatively high in the south of the uranium tailings pond and its adjacent area, while the soil pH in the horizontal profile of 5–15 cm was not significantly different. From the vertical profiles, a downward trend in pH was observed with the increasing soil depth. From the changes in soil EC, in the horizontal profile of 0–5 cm, soil EC was relatively high in the east and center of the uranium tailings pond and its adjacent area, while in the horizontal profile of 5–15 cm, soil EC was relatively high in the whole study area except for a small area. In the vertical profile, the EC of soil increased as the soil depth increased. From the TN content of the soil in the study area, the TN content in the horizontal and vertical profiles was low, and there was no uneven distribution in general except for sporadic areas in the uranium tailings pond. Analysis of the horizontal and vertical profiles indicated a higher soil TOC content in a small area to the southeast of the tailings pond, which showed a downward trend with the increase in soil depth. From the horizontal profiles, the area with a relatively high soil TP content in the horizontal profile of 0–5 cm was only sporadically distributed in the uranium tailings pond, while that area in the horizontal profile of 5–15 cm were distributed in the northwest and south of the study area. Based on the vertical profiles, the area with a relatively high TP content in soil expanded with increasing soil depth.

As shown in Figure 4, from the horizontal profile, the As content of soil in the horizontal profile of 0–5 cm was obviously evenly distributed and was relatively low in the study area, while in the horizontal profile of 5–15 cm, the area with relatively high As content of soil appeared in the northwest of the study area, that was, outside the uranium tailings pond. From the vertical profiles, the area with relatively high As content in soil showed a trend of expanding with increasing soil depth. From the analysis of horizontal and vertical profiles, the area with a relatively high Cd content in soil had a sporadic distribution in the uranium tailings pond, decreased with the increase in soil depth, and tended to be evenly distributed in the horizontal profile of 5–15 cm. The area with a relatively high Pb content in soil in the horizontal profile of 0–5 cm appeared in the east of the uranium tailings pond and its adjacent area, while in the horizontal profile of 5–15 cm, that area appeared in the east and center of the whole study area. In the vertical profiles, the area with a relatively high Pb content in soil showed a trend of increasing with increasing soil depth. From the analysis of horizontal and vertical profiles, the area with a relatively high Zn content in soil in the horizontal profile of 0–5 cm was distributed sporadically in the study area and showed a decreasing trend with increasing soil depth. From the analysis of horizontal and vertical profiles, the Pb and Cr contents in soil in study area had basically the same distribution and change in the horizontal and vertical profiles.

### 3.4. Correlation Analysis

The physicochemical properties and heavy metal/metalloid contents, as well as the previously published radionuclide activity concentrations of soil in the study, were collectively referred to as soil environmental factors [19]. The correlations among the physicochemical properties, heavy metal/metalloid contents, and radionuclide activity concentrations in soil were analyzed by means of the Spearman correlation coefficient. The statistical significance of the Spearman correlations was also calculated by the built-in test method [47].

The visual correlation heatmap among the physicochemical properties, heavy metal/metalloid contents, and radionuclide activity concentrations in soil is shown in Figure 5. As shown in Figure 4, a significant positive correlation ($0.01 < p < 0.05$) between the Zn content and pH in soil was found, which is consistent with the phenomenon described in the literature, that is, in the solutions of neutral and alkaline soil substrates, the concentration of soluble Zn ions is lower than that in light acid soil, which meant that the solubility of Zn ions would decrease with the increase in pH in the soil substrate and could also explain the significant negative correlation ($0.01 < p < 0.05$) between the Zn content and EC in soil [48,49]. The significant positive correlation ($0.01 < p < 0.05$) between the Zn content and

TP content in soil showed that the immobilization of Zn in soil could be highly affected by phosphorus [48].

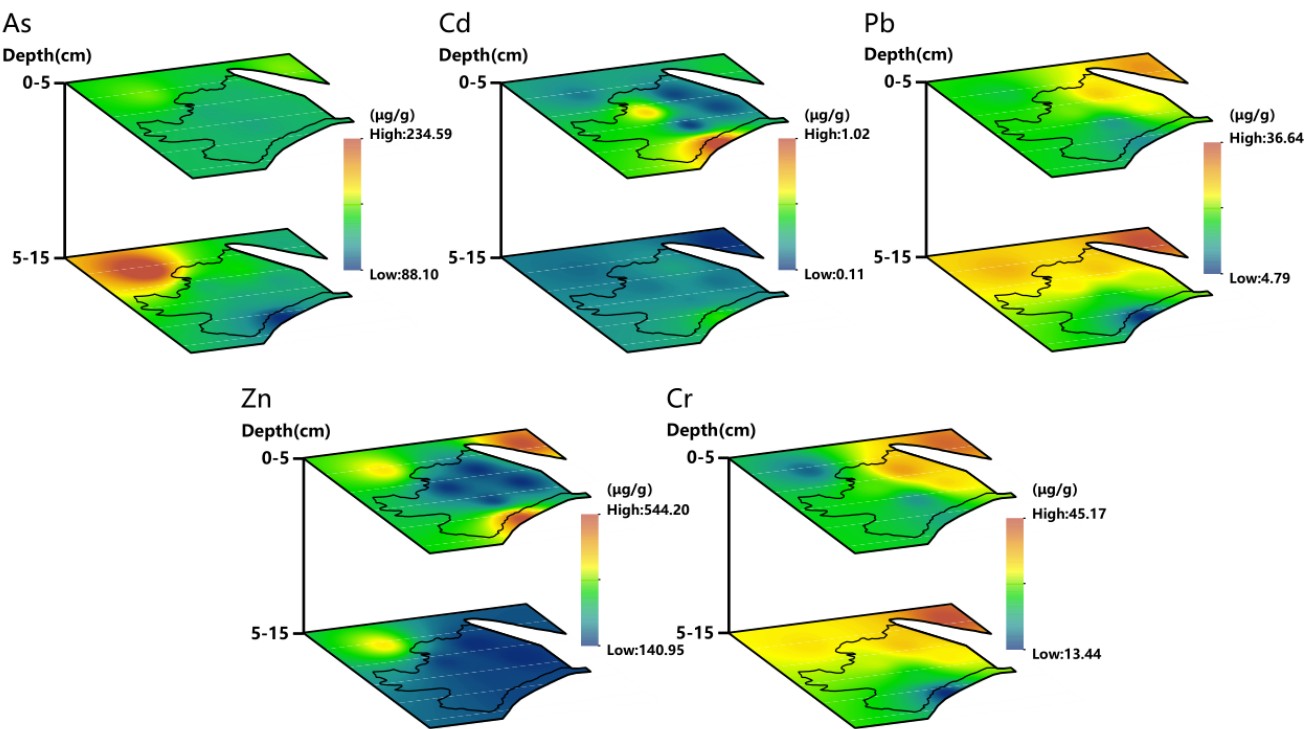

**Figure 4.** Spatial distribution of soil metalloid As and metals (Cd, Pb, Zn, Cr).

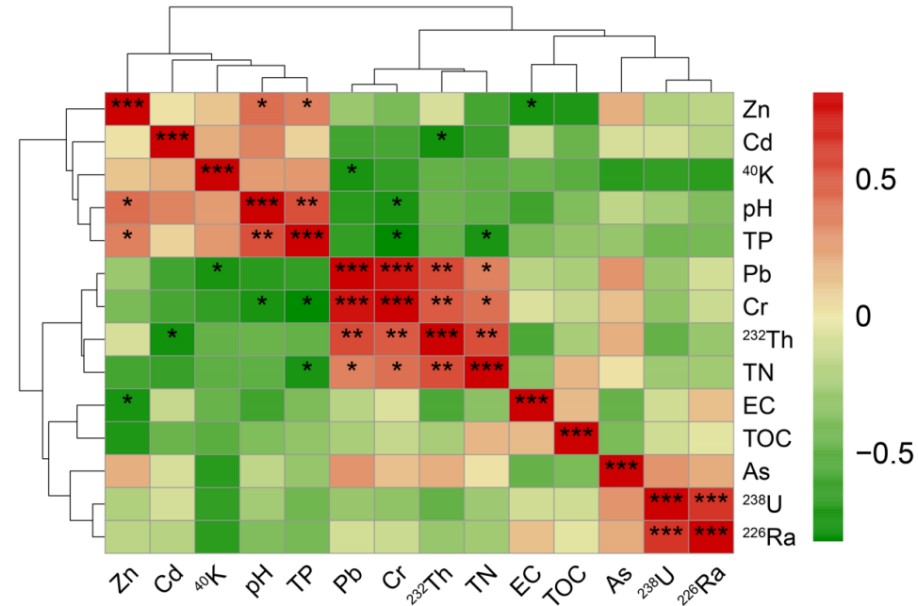

**Figure 5.** Correlation cluster heat map of soil physicochemical properties, metals/metalloids, and radionuclides,. * $0.01 < p < 0.05$, ** $0.001 \leq p < 0.01$, *** $p < 0.001$.

The significant negative correlation ($0.01 < p < 0.05$) between the Cd content and $^{232}$Th activity concentration in soil could be explained that although they are lithophilic elements with similar geochemical properties, CdOH$^+$ species are more likely to occur with the increase in pH in alkaline soil [49].

The Pb content had a significant positive correlation with the TN content ($0.01 < p < 0.05$), [232]Th activity concentration ($0.001 \leq p < 0.01$), and Cr content ($p < 0.001$) in soil, while it had a significant negative correlation with [40]K ($0.01 < p < 0.05$). In soil, $Pb^{2+}$ reacts readily with neutral salts, such as $Sr(NO_3)_2$, $NH_4NO_3$, etc., to form some Pb salts, mainly $Pb(NO_3)_2$ [49,50]. At the same time, $Pb^{2+}$ readily forms organic metal complexes with organic ligands such as glycine and other organic nitrogen forms [51]. Therefore, these could explain the significant positive correlation between the Pb content and TN content in soil. The significant positive correlation between the Pb content and [232]Th activity concentration in soil could be related to secondary lead ([208]Pb) and the decay progeny of radionuclide [232]Th [49,52]. The significant positive correlation between the Pb content and Cr content in soil may be because they were weathered and originated from parent rocks containing $PbCrO_4$ [49]. The geochemical characteristic of $Pb^{2+}$ was similar to the divalent alkaline-earth group of metals, and this element has the ability to replace ions such as K ([40]K in the soil sample ($Bq \cdot kg^{-1}$) = total K in the soil sample ($g \cdot kg^{-1}$) $\times$ 27.9 $Bq \cdot g^{-1}$), Ba, Sr, etc., which could be the reason for the significant negative correlation between the Pb content and [40]K activity concentration in soil [49,53].

Kabata-Pendias (2011) emphasized that $Cr^{3+}$ was almost completely precipitated at pH > 5.5, and its compounds in soil were very stable. However, $Cr^{6+}$ is very unstable and easy to mobilize in alkaline soil, especially in alkaline soil with pH > 8, and the solubility increases with the increase in pH. These observations could be used to explain the significant negative correlation ($0.01 < p < 0.05$) between the Cr content and the pH in soil of the study area [48,49,54]. The significant negative correlation ($0.01 < p < 0.05$) between the Cr content and TP content in soil could be explained as follows: at high pH, the adsorption of phosphorus by oxides is stronger than that of $Cr^{6+}$ [55]. The significant positive correlation ($0.001 \leq p < 0.01$) between the Cr content and [232]Th activity concentration in soil could be explained as follows: they are lithophilic elements with similar geochemical properties [49]. The significant positive correlation ($0.01 < p < 0.05$) between the Cr content and TN in soil could be explained by the pH values (5.5–9.0); $Cr^{6+}$ readily reacts with nitrate and nitrite with the participation of iron ions [49,56]. No significant correlation was found among the As content, physicochemical properties, heavy metals, and radionuclide activity concentrations of soil in this study area.

The significant positive correlation ($0.001 \leq p < 0.01$) between the [232]Th activity concentration and TN content in soil could be explained as follows: Th and nitrate readily react [57]. There was a significant strong positive correlation ($p < 0.001$) between the soil radionuclides [238]U and [226]Ra in the study area since [238]U decays into [226]Ra, and they are both in the same decay sequence [13,19].

The significant negative correlation ($0.01 < p < 0.05$) between the TN content and TP content in soil could be explained as follows: nitrogen and phosphorus are essential elements to maintain life processes and participate in geochemical processes, and microorganisms could play a key role in driving and mediating the nitrogen and phosphorus cycles [58,59].

According to the cluster analysis, the soil pH, TP, radionuclide [40]K, and heavy metals Cd and Zn formed a cluster successively, which could be related to the similar structures and physicochemical properties of Cd and Zn, while Pb, Cr, the radionuclide [232]Th, TN, EC, TOC, metalloid As, and radionuclides [238]U and [226]Ra formed another cluster of the cluster tree, which are basically lithophilic elements with similar geochemical properties [49].

## 4. Conclusions

In this paper, the determination of soil physicochemical properties and heavy metals/metalloids at two different profiles in the tailings pond and its adjacent area was completed, and the soil physicochemical properties and the spatial distribution characteristics of metals/metalloids in the study area were analyzed. The conclusions were as follows: (1) the physicochemical properties (pH, EC, TN, TOC, and TP), heavy metal (Cd, Pb, Zn, Cr) contents, and metalloid (As) content of the different soil profiles (0–5 cm, 5–15 cm) in

the tailings pond and its surrounding areas were within the background value range of China and other countries or regions around the world; (2) the visual spatial distribution map showed that the spatial distribution characteristics of the EC, TP content, Pb content, and Cr content in the soil in the study area increased with the increase in depth of the vertical profile; (3) there were significant positive correlations among heavy metals and radionuclides and significant negative correlations among heavy metals, radionuclides, and physicochemical properties; (4) the cluster tree divided the environmental factors into two clusters; pH, TP, $^{40}$K, Cd, and Zn formed one cluster, and Pb, Cr, $^{232}$Th, TN, EC, TOC, As, $^{238}$U, and $^{226}$Ra formed another cluster. Based on the above analysis, the uranium tailings pond is in good operation, and no migration and diffusion of heavy metals/metalloids to the surrounding soil ecological environment were found.

**Author Contributions:** Conceptualization, Y.M. and J.Y.; Data curation, J.Y.; Funding acquisition, G.F.; Investigation, B.W., H.C. and Y.H.; Methodology, Q.L.; Writing—original draft, Y.M., J.Y. and Q.L.; Writing—review & editing, Y.M., J.Y., B.W., H.C., Y.H. and G.F. All authors have read and agreed to the published version of the manuscript.

**Funding:** The work presented here was performed under the auspices of the National Natural Science Foundation of China, Project No. 32060292.

**Institutional Review Board Statement:** Not applicable.

**Informed Consent Statement:** Not applicable.

**Data Availability Statement:** The original data of this paper were mainly obtained through the authors' experiment.

**Acknowledgments:** For providing facilities and support, the authors are grateful to the Radiation Environment Supervision Station of Xinjiang, China, and the Urumqi Customs Technology Center, China.

**Conflicts of Interest:** There was no conflict of interest reported by the authors.

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
