# Peer review of "Heavy Metals/Metalloids in Soil of a Uranium Tailings Pond in Northwest China: Distribution and Relationship with Soil Physicochemical Properties and Radionuclides"

_sustainability, doi:10.3390/su14095315_

Round 1

Reviewer 1 Report

This study researched the distribution characteristics of several heavy metals/metalloids in the soil of a uranium tailing pond in Northwest China and discussed their relationship with soil physicochemical properties. In general, the results obtained from this research are meaningful for reasonably assessing and predicting the environmental risk of heavy metals/metalloids in uranium tailing areas. I recommend this paper can be published in the Sustainability journal after a necessary revision. Some specific comments are listed as follows:

  • In the introduction, the author had better give some specific examples or data to highlight the seriousness of heavy metal/metalloid pollution in uranium tailing pond.
  • Specific location (latitude and longitude information) of the study site should be given.
  • A distance marker must be placed in Figure 1.
  • “2.3 Analytical methods”, the name and model of the instruments for measuring the concentration of heavy metal/metalloid should be given.
  • Page 4, which type of soil in the LUCAS report was used for comparison?
  • Page 7, the heavy metal/metalloid concentrations of the soil in the study sites seemed to be relatively low, indicating the pollution was not serious?
  • Only results are presented in the 3.3 section, appropriate discussion should be given.
  • Environmental significance reflected from this study should also be discussed.
  • The conclusion should be more concise and general.
  • There are many grammar mistakes in the paper, the language of the paper needed to be thoroughly improved.

Author Response

Dear reviewer,

Reviewer 2 Report

The work of Yu Mao, Jinlong Yong, Qian Liu, Baoshan Wu, Henglei Chen, Youhua Hu, Guangwen Feng “Heavy metals/metalloids in soil of a uranium tailings pond in Northwest China: Distribution, relationship with soil physico-chemical properties and radionuclides” was carried out at a high scientific and methodological level. The data and statistical calculations and hypotheses are carefully verified and reliable. The work is worthy of publication in “Sustainability” journal.

There are only a few comments, mainly of a technical nature:

- Page 2, Item 2.1. Study area: “where the uranium tailings pood (about 20,000 m2)…” – may be “pond”?

- Page 3, Item 2.2. Sampling and sample pretreatment: “The dried soil samples were ground into a fine powder with a test sieve the powder samples passing 2 mm aperture sieve were used for determination of soil pH and electrical conductivity (EC), while those passing 0.15 mm aperture sieve were used for soil total nitrogen (TN), total organic carbon (TOC), total phosphorus (TP) and heavy metal/metalloid.” – a fine powder?

- Page 4, Item 3.1. Soil physicochemical properties: “According to the analysis of the measured pH values, the measured values of pH in the study area were within the range of domestic soil background value of pH: 3.10-10.60 [28],…” - “According to the analysis of the measured pH values” - unnecessary part of the sentence.

- Page 9, Item 3.4. Correlation analysis: “The Pb content had a significant positive correlation with TN content (0.01<P<0.05), 232Th activity concentration (0.001≤P<0.01), and Cr content (P < 0.001) in soil, while it had a significant negative correlation with 40K (0.01<P<0.05). In soil, Pb2+ was easy to react with neutral salts, such as Sr(NO3)2, NH4NO3, etc., to form some Pb salts mainly Pb(NO3)2. Therefore, it could be used to explain the significant positive correlation between Pb content and TN content in soil [46,47].” - Such a statement about the reason for the positive correlation between Pb and TN is not entirely justified, since the authors did not provide data on the content of organic nitrogen in the studied soils.

Author Response

Dear reviewer,

Round 2

Reviewer 1 Report

The manuscript has been well revised and it is acceptabble now.